# Statistically valid explainable black-box machine learning: applications in sex classification across species using brain imaging

Tingshan Liu[1], Jayanta Dey[1], Beiya Xu[1], Eric W. Bridgeford[2], Samuel Alldritt[3], Karl-Heinz Nenning[4], Kyoungseob Byeon[3], Ting Xu[3], Joshua T. Vogelstein[5]*

1 Department of Biomedical Engineering, Johns Hopkins University, Baltimore, Maryland, United States of America, 2 Department of Psychology, Stanford University, Stanford, California, United States of America, 3 Center for the Developing Brain, Child Mind Institute, New York, New York, United States of America, 4 Nathan S. Kline Institute for Psychiatric Research, Orangeburg, New York, United States of America, 5 Department of Biomedical Engineering, Institute for Computational Medicine, Kavli Neuroscience Discovery Institute, Johns Hopkins University, Baltimore, Maryland, United States of America

☯ These authors contributed equally to this work.
* jvogles3@jh.edu

## Abstract

Sex classification using neuroimaging data has the potential to revolutionize personalized diagnostics by revealing subtle structural brain differences that underlie sex-specific disease risks. Despite the promise of machine learning, traditional methods often fall short in providing both high classification accuracy and interpretable, statistically validated feature importance scores for high-dimensional imaging data. This gap is particularly evident when conventional techniques such as random forests, LIME, and SHAP are applied, as they struggle with complex feature interactions and managing noise in large datasets. We address this challenge by developing an integrated framework that combines Oblique Random Forests (ORFs) with a novel, permutation-based feature importance testing algorithm. ORFs extend traditional random forests by employing oblique decision boundaries through linear combinations of features, thereby capturing intricate interactions inherent in neuroimaging data. Our feature importance testing method, NEOFIT, rigorously quantifies the significance of each feature by generating null distributions and corrected $p$-values. We first validate our approach using simulated datasets, establishing its robustness and scalability under controlled conditions. We then apply our method to classify sex from both voxel-wise structural MRI and cortical thickness data in humans and macaques, facilitating direct cross-species comparisons. ORFs achieves AUC > 0.80 on human data, and >0.70 on macaque data, while NEOFIT identifies statistically significant features aligned with sex-dimorphic neuroanatomy. Our results demonstrate that the proposed framework not only enhances classification performance but also provides clear, interpretable insights into the neuroanatomical features that distinguish sexes. These methodological advancements pave the way for improved diagnostic tools and

**Data availability statement:** All relevant links to the data are within the manuscript and its Supporting Information files.

**Funding:** This work was supported by the National Institutes of Health (https://www.nih. gov) under award numbers RF1MH128696 and U19NS104653, and by the National Science Foundation (https://www.nsf.gov) under award number NSF 20–540. The funders had no role in study design, data collection and analysis, decision to publish, or preparation of the manuscript.

**Competing interests:** The authors have declared that no competing interests exist.

contribute to a deeper understanding of the evolutionary basis of sex differences in brain structure.

## Introduction

The opportunity to harness high-dimensional medical imaging data to improve diagnostic precision and treatment outcomes is immense. Prior studies in radiography and ultrasound have reported clinically relevant diagnostic performance from Machine Learning-based pipelines [1,2]. In neuroimaging, structural Magnetic Resonance Imaging (MRI) measurements offer a wealth of information that can reveal subtle brain differences linked to neurodevelopmental, cognitive, and behavioral traits, as well as susceptibility to disorders such as autism, schizophrenia, and Alzheimer's disease [3–9]. To realize this promise in clinical settings, methods are needed that not only deliver high predictive accuracy but also provide interpretable, statistically validated insights into which features are truly significant.

A major challenge in neuroimaging classification is ensuring biological interpretability. Feature selection and rigorous statistical validation are crucial when models handle thousands of features. While popular post-hoc explanation techniques such as Local Interpretable Model-Agnostic Explanations (LIME) and Shapley Additive Explanations (SHAP) offer valuable insights [10,11], they face challenges in high-dimensional settings. LIME relies on local linear approximations, which can lead to variability in the explanations when the feature space is large and complex. Similarly, although SHAP's Shapley values are theoretically robust, their computation in high-dimensional data can be computationally intensive and may result in noisy estimates. A critical gap remains in the absence of a robust method that consistently discriminates genuine signal from noise in high-dimensional neuroimaging data.

We address this gap by developing an integrated framework that combines Oblique Random Forests (ORFs) with a permutation-based feature importance testing algorithm. ORFs, which use linear combinations of features at each split [12], are particularly adept at capturing complex interactions inherent in neuroimaging data. Our feature importance testing framework statistically validates feature contributions by generating null distributions and applying Bcorrected *p*-values, thereby reducing overfitting and enhancing interpretability.

To systematically evaluate our approach, we first benchmark it on simulated datasets to assess its effectiveness in feature selection under controlled conditions. We then apply it to human and macaque neuroimaging data to classify biological sex using both voxel-wise MRI features and parcellated cortical thickness (CTh) data.

This workflow allows us to validate our method's robustness before interpreting real neuroanatomical differences across species. Our results highlight both conserved and species-specific patterns of sex-related structural variation, with key differentiating regions emerging in the limbic system, sensory processing areas, and motor-associated regions. By integrating machine learning with statistical validation, our framework provides an interpretable and scalable approach for neuroimaging classification, with broader applications beyond sex classification.

## Materials and methods

### Overview

To evaluate our classification framework, we first apply it to simulated datasets, providing a controlled benchmark for assessing feature selection and classification accuracy. This establishes the effectiveness of our approach before transitioning to structural MRI data. We then analyze human and macaque MRI datasets, using oblique random forests and feature importance testing to classify biological sex. MRI preprocessing ensures cross-species comparability, with both structural MRI volume and cortical thickness data serving as primary feature spaces for classification. This two-stage workflow—benchmarking on simulations followed by real-data application—allows a rigorous assessment of our method's reliability in identifying key brain regions relevant to sex classification.

### Oblique random forest

We employed Oblique Random Forests (ORF) for classification tasks, specifically leveraging two variants from the `tree-ple` library: Sparse Projection Oblique Random Forest (SPORF) [13] and Manifold Oblique Random Forest (MORF) [14]. These methods are beneficial for addressing high-dimensional data with complex relationships, as they extend traditional axis-aligned decision trees by allowing each decision tree to split data using random projections, enhancing the ability to capture non-linear patterns and interactions within the feature space. ORFs are particularly advantageous in neuroimaging analysis, where data are often high-dimensional, and the relationships between features are non-linear and complex and may not align with traditional grid-like partitions of the data.

We specifically selected SPORF and MORF for their suitability in handling cortical thickness and MRI volume data, respectively. SPORF has advantages over other oblique random forests primarily because it integrates sparsity-inducing regularization [13]. This feature is particularly beneficial when working with datasets that may contain a large number of irrelevant or redundant features, as it helps to focus the model on the most important variables. For our cortical thickness data, SPORF's ability to select a subset of relevant features improves interpretability and reduces overfitting, which is crucial for our sex classification task. On the other hand, MORF was chosen for MRI volume data due to its robust handling of correlated features [14]. Since MRI data may have intricate spatial correlations between features, incorporating feature locality could improve model performance by capturing these spatial dependencies. Both SPORF and MORF handle high-dimensional data efficiently, making them well-suited for the relatively high-dimensional nature of our data [13,14].

To validate model performance, we used the out-of-bag (OOB) score, which provides an internal validation metric. The OOB score is computed by evaluating each tree on the data points that were not used during its construction (i.e., those not included in the bootstrap sample). This technique helps estimate the generalization ability of the model without requiring a separate validation dataset. The OOB score is particularly useful in neuroimaging studies, where cross-validation can be computationally expensive. Using OOB scoring, we were able to efficiently assess model performance and avoid overfitting by providing a robust estimate of classification accuracy. S1 Fig depicts a flow diagram for our classification framework.

### Feature importance testing

Feature importances from Random Forest (RF) provide insights into how strongly each feature contributes to the model's predictions [15]. In RFs, feature importances are typically computed using the Gini impurity-based measure: each feature's importance is proportional to the total reduction in Gini impurity it achieves when used to split nodes across all trees in the forest. For SPORF, this process is enhanced by the sparse projection mechanism, where only a subset of features is considered at each split, ensuring that the feature importance estimates focus on the most relevant variables while mitigating noise from irrelevant ones. In MORF, feature importances are computed similarly but are influenced by the method's consideration of feature locality, allowing it to capture spatial correlations in MRI volume data. This leads to a more nuanced understanding of how spatially distributed voxel values contribute to classification tasks.

To further contextualize the feature importance analysis, we compare our approach with two popular model-agnostic algorithms for feature importance: Local Interpretable Model-Agnostic Explanations (LIME) and Shapley Additive Explanations (SHAP). LIME, introduced by Ribeiro et al. [10], aims to explain individual predictions by approximating the complex model locally with a simpler, interpretable model. It computes feature importances by perturbing the input data and observing the changes in model predictions, providing insight into the local behavior of the classifier. SHAP, on the other hand, is grounded in cooperative game theory and computes feature importance by calculating the Shapley values, which represent the average contribution of each feature across all possible permutations of feature combinations [11]. Both LIME and SHAP are widely adopted in machine learning for interpreting black-box models and provide global and local feature importance measures. While these methods are powerful, they can be computationally expensive and may not scale well to high-dimensional neuroimaging data. Our feature importance testing algorithm, which adapts the tree permutation method, offers a more scalable solution for assessing the statistical significance of features in such high-dimensional settings.

To evaluate the significance of feature importances, we adapt and extend the tree permutation method proposed by Coleman et al. [16] to address the unique challenges of high-dimensional neuroimaging data. This method enables scalable and efficient hypothesis testing in random forests by generating null distributions of feature importance values through tree permutations [16]. Rather than generating $n$ independent forests for permutation testing, it constructs a null distribution by randomly shuffling trees within an ensemble, preserving the dependency structure while drastically reducing computional overhead. To make it feasible for neuroimaging, we modify the framework by leveraging a two-forest approach: one baseline forest and another with permuted features. Additionally, we incorporate Bonferroni-Holm correction to control for multiple comparisons across millions of features, ensuring robust significance assessment. This allows us to assess which cortical thickness or voxel features have statistically significant contributions to sex classification. The details of this method are outlined in Algorithm 1, Neuro-Explainable Optimal Feature Importance Testing (NEOFIT).

## Algorithm 1: Training an Explainable Classifier using NEOFIT: Neuro-Explainable Optimal Feature Importance Testing.

**Input:** Dataset $\mathcal{D}$ with features and labels, number of trees $n_{\text{trees}}$, number of permutations $n_{\text{perm}}$, number of bootstraps $B$.
**Output:** Feature importance scores **I**, adjusted $p$-values.

```
1:   for b ∈ B do
2:       𝒟_b ← Bootstrap(𝒟)
3:       I_b ← FitDecisionTree(𝒟_b)              ▷ Train tree, compute feature importances
4:       𝒟'_b ← ShuffleLabels(𝒟_b)
5:       I'_b ← FitDecisionTree(𝒟'_b)
6:   end for
7:   function ComputeTestStatistic(I, I')
8:       R ← Rank(I),  R' ← Rank(I')            ▷ Rank importances
9:       T_i += 𝕀(R'_i > R_i) for all i ∈ {1, ..., d}  ▷ Update statistics
10:      T_i ← T_i/n_trees
11:  end function
12:  for p ∈ {1, ..., n_perm} do
13:      for t ∈ {1, ..., n_trees} do
14:          (I_t^perm, I'_t^perm) ← PermutePairs(I_t, I'_t)
15:      end for
16:      T_p^null ← ComputeTestStatistic(I^perm, I'^perm)   ▷ Compute null statistic
17:  end for
```
18:  $p_i \leftarrow \frac{1 + \sum_{p=1}^{n_{\text{perm}}} \mathbb{I}(T_i^{\text{null}} > T_i)}{1 + n_{\text{perm}}}$  for all $i$    ▷ Compute p-values
```
19:  P ← Bonferroni-Holm(p)
```
- Bootstrap($\mathcal{D}$): Draw a bootstrap sample from dataset $D$.
- FitDecisionTree($\mathcal{D}$): Train a decision tree on sample and compute feature importances.
- ShuffleLabels($\mathcal{D} = (X, y)$): Apply a random permutation such that $y' = \sigma(y)$.

- RANK(**I**): Rank feature importances **I** from the largest. Assign rank d to missing features.
- PERMUTEPAIRS(**I, I'**): Shuffle the feature importance values in **I** and **I'** together while maintaining their pairwise associations.
- BONFERRONI-HOLM(**p**): Apply the Bonferroni-Holm correction to $p$-values **p**.

## Simulated datasets

To demonstrate the efficacy of our feature importance testing algorithm, NEOFIT, we performed simulations using the MNIST and Trunk datasets. These simulations allowed us to evaluate the algorithm's performance in controlled settings before applying it to real-world human and macaque MRI data.

For the MNIST dataset, we followed the setup described in Li et al. [14], specifically Fig 5, where the goal was to classify digits 3 and 5 out of the full MNIST set. The MNIST dataset contains images of handwritten digits (28 × 28 pixels), resulting in a high-dimensional feature space of 784 features per image. In our simulation, we used a subset of the dataset with only 3 and 5 for a binary classification task. This allowed us to test our algorithm's ability to identify significant features in a relatively simple, yet high-dimensional, dataset. The MNIST dataset is commonly used to benchmark classification methods, especially when assessing the performance of algorithms designed to handle nonlinearities in data.

For the Trunk simulation, we replicated the setup from Tomita et al. [13], specifically Fig 4, which involves comparing two Gaussian distributions. Both distributions have an identity covariance matrix, with the first distribution having a mean vector starting at −1 and decreasing by a factor of $\sqrt{i}$ for each dimension $i$, while the second distribution's mean vector is the negative of the first. As the dimensionality $d$ of the data increases, the two distributions become closer, making the task of classification progressively more difficult. The mathematical formulation of these two distributions is provided in S1 Appendix. The Trunk dataset is specifically designed to simulate high-dimensional, sparse data and tests the ability of algorithms like SPORF to efficiently identify relevant features in a sparse and increasingly challenging classification task. The simplicity of the Trunk dataset makes it an ideal setting for evaluating the feature importance and selection abilities of our algorithm in a controlled, synthetic environment.

Both of these simulations allowed us to rigorously test the performance of our feature importance testing Algorithm in scenarios that mimic real-world challenges in feature selection and classification. By applying this method to both the MNIST and Trunk datasets, we can demonstrate its ability to reduce computational overhead while maintaining classification accuracy, showcasing its suitability for handling high-dimensional neuroimaging data.

## Simulated performance

In our simulation experiments, we aimed to compare the performance of four feature selection algorithms—RF, LIME, SHAP, and NEOFIT—across two datasets: MNIST, a widely used benchbark dataset, and Trunk, a simulated dataset. The main objective was to evaluate how effectively each algorithm selects the most informative features for classification tasks when varying the number of features, ranging from 1 to 512 features.

For each feature dimension $d$, the dataset was partitioned into three subsets: training (60%), validation (20%), and testing (20%). In each iteration, we trained a RF on the training set and ranked the features by their importance using the Gini impurity measure. Then, we selected the top $d$ features based on this ranking, trained a new RF on the validation set using only these features, and evaluated the resulting model on the testing set. This procedure was repeated for 50 iterations to ensure stability and robustness in the results.

We also tested the performance of LIME, SHAP, and our feature importance testing algorithm on the same feature importances derived from RF. LIME and SHAP are popular techniques for model interpretability: LIME approximates the model locally, and SHAP uses Shapley values to attribute contributions to individual features. Our algorithm, NEOFIT, applied the tree permutation method to generate null distributions of feature importances and selected features based on statistically significant $p$-values, providing a more rigorous approach to feature selection.

In both MNIST and Trunk simulations, our algorithm consistently outperformed the others. In the MNIST dataset, we achieved over 97% accuracy using fewer features than any other method, demonstrating the efficiency of our approach (Fig 1). For the Trunk dataset, we obtained over 80% accuracy, which was higher than the best performance of all other algorithms, even with a relatively small number of features (Fig 1). These results underscore the superiority of our method, particularly in high-dimensional data, where both accuracy and feature efficiency are crucial.

## Structural MRI dataset

The human MRI volume data were sourced from publicly available repositories, including the 1000 Functional Connectomes (FCON1000), Healthy Brain Network (HBN), and several other initiatives. A full list of the datasets can be found in S1 Table in the supplementary materials. We included data from unrelated participants across multiple cohorts to ensure diversity and replicability. Specifically, T1-weighted anatomical MRI data were analyzed. Detailed preprocessing protocols, as well as demographic and acquisition parameters, can be found on the respective dataset portals. There are 14,380 subjects in total, with 7,416 females and the remaining 6,964 males, having a mean age of 39 for females and 31 for males. S2 Fig provides the full age and sex distribution for human data.

The Non Human Primate (NHP) MRI data were sourced from the PRIME-DE consortium [17] and included 592 rhesus macaques (*Macaca mulatta*), including 265 females and 327 males, scanned under anesthesia. Throughout this manuscript, NHP and macaque are used interchangeably to refer to rhesus macaques. The mean age is 2.0 for females and 1.8 for males. Final volumes were resampled to match the resolution of the human data, facilitating direct comparisons across humans and macaques. For further details, please refer to the source of the data at PRIME-DE: UW-Madison. Detailed age and sex distributions for macaque data are also shown in S2 Fig.

## MRI preprocessing

The MRI preprocessing pipeline was designed to ensure high-quality input data for subsequent analyses. Following established preprocessing protocols [18], the raw T1-weighted images underwent several key steps, including denoising, brain extraction, and tissue segmentation. First, noise reduction was applied to enhance image quality while preserving anatomical details. Brain extraction (or skull stripping) was then performed to remove non-brain tissues, facilitating accurate

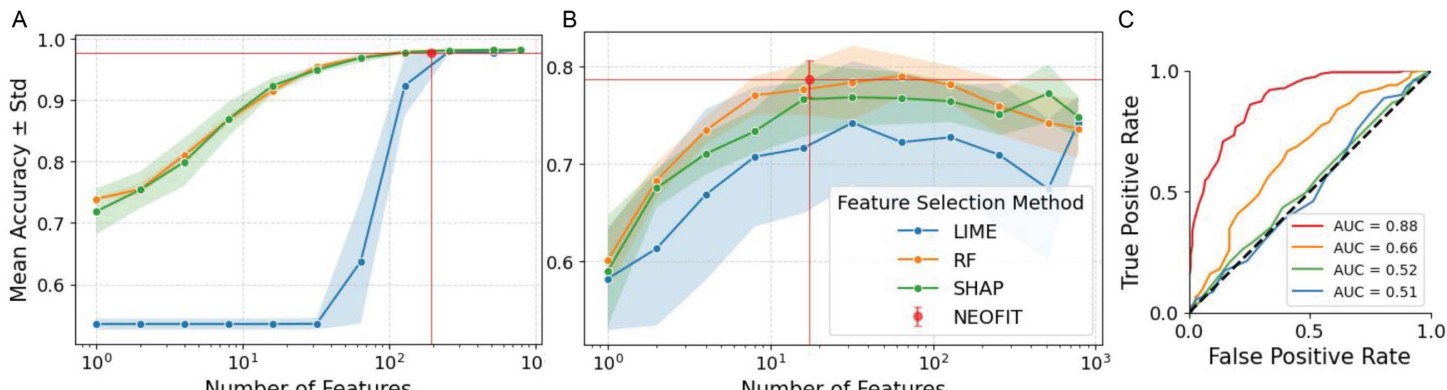

**Fig 1. Simulation results for feature selection performance.** A: Classification accuracy (mean ± standard deviation) for MNIST digits 3 vs. 5 across different numbers of selected features. B: Results for the Trunk dataset. 5 repetitions were run per condition. Our method, NEOFIT, is indicated by the red dot, with red horizontal and vertical lines marking the corresponding accuracy and feature count. C: ROC curve for Trunk simulation including $n = 17$ top features where $n$ is determined by NEOFIT.

cortical and subcortical measurements. The images were subsequently segmented into different tissue classes, such as gray matter, white matter, and cerebrospinal fluid.

To ensure data reliability, an initial quality control step was conducted to screen for artifacts, motion-related distortions, or structural anomalies. Preprocessed data were further reviewed through visual inspection, and scans that failed at any stage of processing were either corrected or excluded from the final dataset.

To generate gray and white matter density maps, we employed voxel-based morphometry (VBM) using the Statistical Parametric Mapping (SPM) framework. For human data, preprocessing was performed using CAT12, an advanced and optimized pipeline for high-resolution human brain imaging [19]. The CAT12 toolbox enabled improved tissue segmentation and bias field correction, ensuring robust gray and white matter probability maps [19]. For macaque data, a customized template was used in conjunction with methods adapted from SPM-mouse [20]. This approach accounted for structural differences between human and macaque brains, improving segmentation accuracy and ensuring valid cross-species comparisons.

### Cortical thickness dataset

Cortical thickness (CTh) measurements were derived from surface-based morphometric analyses tailored for both human and macaque brains. For human subjects, CTh was extracted using the CAT12 toolbox, an advanced framework optimized for surface reconstruction and thickness estimation [19]. This pipeline includes topology correction, spherical mapping, and projection-based thickness estimation to ensure accurate and reliable measurements. For macaques, CTh was computed using a species-specific pipeline based on a customized surface template, following methodologies from the Macaque CHART framework [18], which improves precision in non-human primate surface morphometry.

To enable cross-species comparisons, CTh measurements were parcellated using an established alignment framework [21], ensuring that homologous cortical regions were systematically mapped between humans and macaques. This approach minimizes species-specific biases and enhances the biological interpretability of interspecies analyses.

CTh data were processed under two parcellation schemes: Schaefer and Markov. The Schaefer parcellation, based on functional connectivity gradients, provides a data-driven, hierarchical atlas designed to balance spatial resolution and functional specificity [22]. We used the 200-region version to optimize computational efficiency while retaining detailed cortical representation. In contrast, the Markov parcellation is derived from anatomical connectivity patterns in macaques, based on tracer studies that define 182 structurally connected cortical regions [23]. While originally designed for different species, both atlases were adapted for cross-species analysis through mapping transformations, enabling direct comparison between human and macaque cortical organization.

The dataset includes CTh measurements from 10,608 human and 572 macaque subjects, each analyzed under both parcellation schemes, yielding four datasets: {Human, Macaque} × {Schaefer, Markov}. These datasets provide a comprehensive framework for investigating species-specific and evolutionarily conserved cortical patterns, facilitating robust statistical analysis of sex-related cortical differences and their biological significance. A detailed depiction of the age and sex distributions for CTh data in both species is provided in S3 Fig.

### Variable specification and relationship identification for within- and cross-species MRI analysis

To make our structural assumptions underpinning our analysis explicit, we employ a directed acyclic graph (DAG) framework [24,25]. DAGs formalize causal relationships between variables and provide graphical criteria for whether particular effects can be identified within the context of a given analysis. Fig 2 illustrates our assumed causal structure, where our primary interest is the effect of biological sex (the *exposure*, green) on brain structural features (the *outcomes*, blue): voxel-wise gray and white matter density maps, or parcellated cortical thickness. The total effect includes pathways mediated by intracranial volume (ICV) and hormonal status (the *mediators*, tan). Since the relationship between ICV and brain structure is well-established and represents a nuisance scaling factor

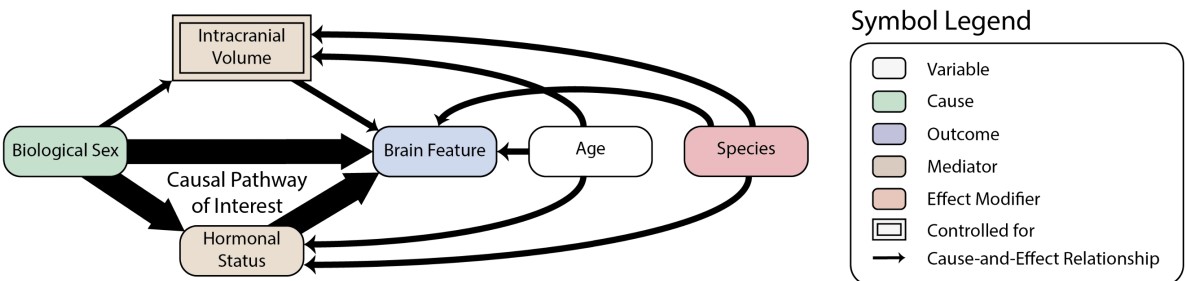

**Fig 2. Directed acyclic graph illustrating the assumed causal structure for sex classification analysis.** Nodes represent variables, with biological sex (green) as the exposure and brain features (blue) as outcomes. Mediators (tan) include intracranial volume (ICV, boxed to indicate that our analysis controls for ICV differences) and hormonal status. Age (white) affects hormonal status and brain features. Species (pink) acts as an effect modifier. Arrows indicate assumed causal relationships.

rather than our primary interest [3], we employed species-specific spatial normalization, which intends to control for ICV differences (indicated by the boxed border in Fig 2). Hormonal status, conversely, represents a key biological mechanism through which sex differences manifest; controlling for hormones would remove the effect we aim to study. Therefore, our within-species analyses seeks to quantify brain regions where biological sex has a significant effect, conditional on ICV normalization but preserving hormone-mediated pathways. Age (white) is a common cause of hormonal status and brain features. Finally, our cross-species analysis examines whether species (the *effect modifier*, pink) modifies the magnitude of sex effects on brain features, testing for evolutionary conservation versus species-specific adaptation.

## Results

### Hyper-parameter tuning

We fine-tuned several hyper-parameters for both SPORF and MORF implemented with `treeple` (https://treeple.ai/):

- `max_features`: Determines the maximum number of features considered for each split. A lower value encourages diversity among trees, preventing overfitting; a higher value can lead to stronger splits and higher accuracy for individual trees while too high a value may lead to less randomization and poorer generalization.

- `max_patch_dim (for MORF):` Defines the dimensionality of the patches used for splitting the data. Adjusting this parameter allows the model to explore different feature subspaces and balance model complexity and accuracy.

- `feature_combinations` (for SPORF): Controls the number of linear combinations of features evaluated at each split in the tree. A higher value leads to a less sparse model, as it incorporates more features into each decision. Increasing the number of combinations improves the flexibility of the model but may reduce interpretability and increase computational complexity.

 Neither `max_features` nor `max_patch_dim` affect the OOB score for MRI volume data (results not shown). For CTh data, `feature_combinations` and `max_features` influence human data far more than macaque data (Fig 3). Given that SPORF applies linear combinations of features at each split, we tested three types of feature normalization and evaluated their impact on classification performance. Based on these comparisons (S2 Fig), we selected z-score normalization as it yielded the most stable and consistent performance across datasets.

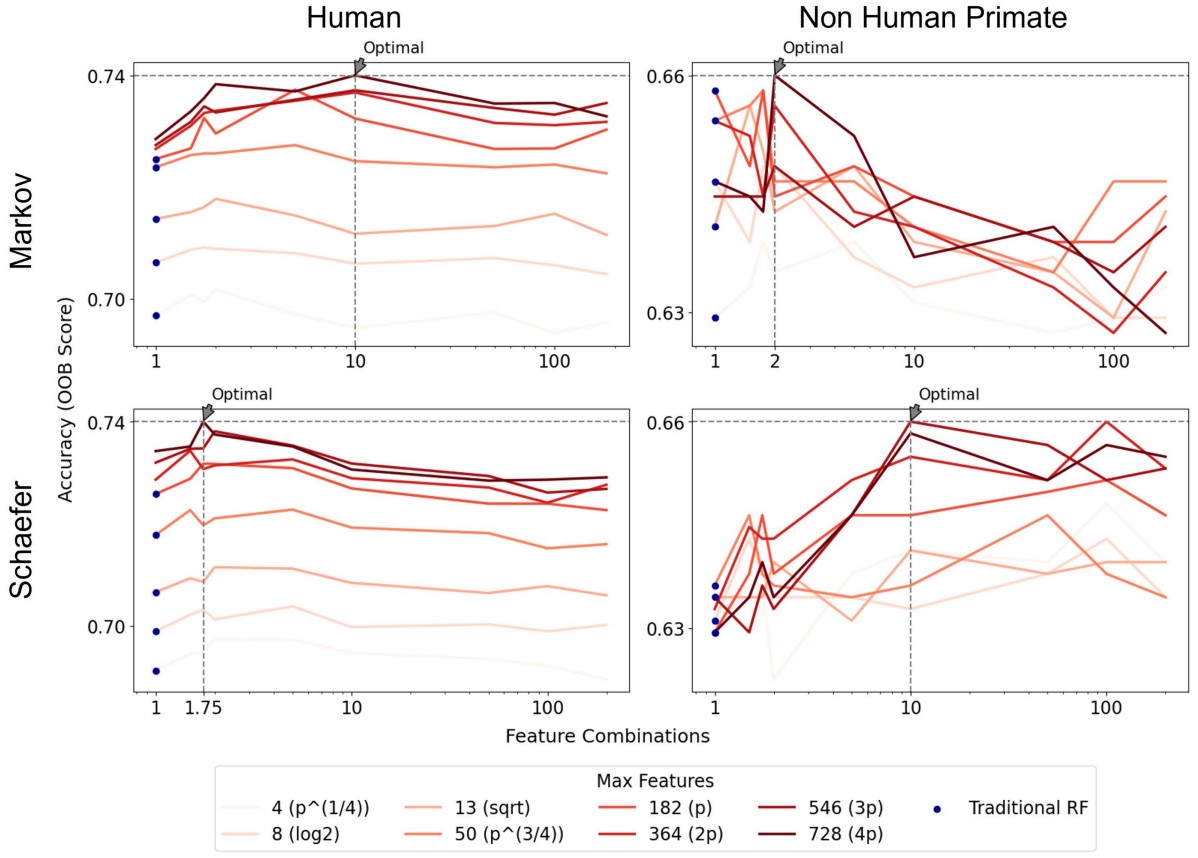

**Fig 3. Performance comparison of oblique random forest (SPORF) using cortical thickness data across various hyper-parameters.** The figure displays the impact of different model configurations, including `feature_combinations`, and `max_features` on performance. Blue dots are achievable by traditional random forest models. In subsequent experiments, we set `max_features` to *4p* for all CTh data. For `feature_combinations`, we used 10 for human Markov, 2 for macaque Markov, 1.75 for human Schaefer, and 10 for macaque Schaefer. The number of estimators was set to 5,000 for human data and 20,000 for macaque data. All other parameters followed the default settings of the `treeple` implementation (https://treeple.ai/).

## Performance of the sex classifier

After hyper-parameter optimization, we evaluated the performance of our random forest models using the area under the curve (AUC) from the receiver operating characteristic (ROC) curve, which quantifies classifier performance across various decision thresholds. Fig 4 illustrates the ROC curves and corresponding AUC scores for both MRI volume and CTh data. Our model demonstrates an especially high AUC for human MRI volume data, while exhibiting moderate classification performance for macaque data. Overall, the human data consistently outperforms the macaque data across both imaging modalities.

## Performance of the feature importantce testing

To compare our method with previous interpretable feature importance testing approaches on the sex-classification task, we replicated the procedure described in the Materials and Methods section on CTh data.

Features were selected using LIME and SHAP applied to trained Random Forests, with the number of trees matched across species (5,000 for human and 20,000 for macaque). We then trained classifiers using only the selected significant features and reported accuracy as a proxy for the reliability of each selection method. For LIME and SHAP, we varied the

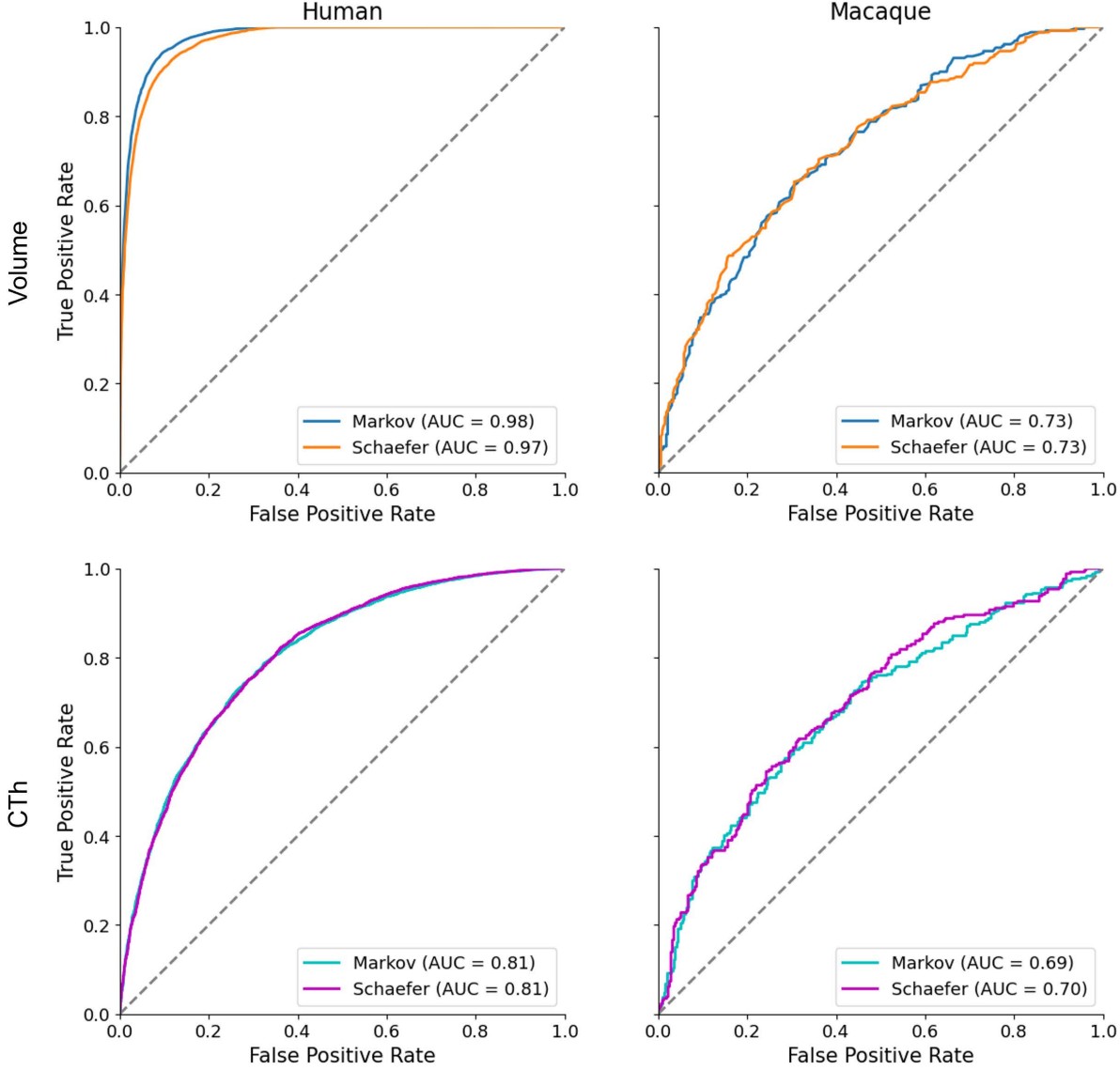

**Fig 4. Comparison of True Positive Rate versus False Positive Rate for sex classification models on MRI volume data (top row) and CTh data (bottom row).** Panels show results for (left) humans and (right) NHPs. Curves are color-coded by tissue type (gray/white matter) for MRI volume data and by parcellation scheme (Markov/Schaefer) for CTh data. The AUC is annotated for each category.

number of top features to examine performance as a function of the selected-feature count; our methods, NEOFIT, yields a single, data-driven estimate of the number of significant features. Fig 5 reports results on the Markov parcellation data; corresponding results for Schaefer parcellation are provided in S5 Fig.

On the human Markov CTh data, NEOFIT reaches around an accuracy of 0.70 with 29 features, whereas LIME requires a comparable subset and SHAP requires a much larger subset to match the same level. On the NHP Markov CTh data, NEOFIT achieves an accuracy of approximately 0.66 using 44 features, which is above the standard deviation bands of LIME and SHAP across most feature-subset sizes. Unlike the post hoc methods, NEOFIT does not require

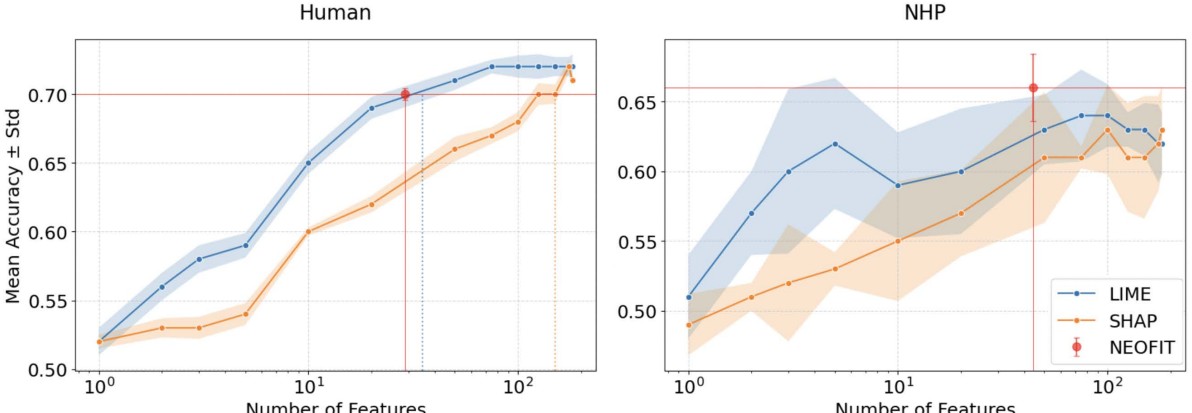

**Fig 5. Comparison of feature selection performance on Markov CTh data.** Classification accuracy on human Markov CTh data (left) and macaque Markov CTh data (right) versus the number of selected features for each method. Mean and standard deviation are computed over five repetitions per condition. NEOFIT is shown as a red point at its chosen feature count.

tuning the feature-subset size and shows lower variance. Taken together, these results indicate that NEOFIT delivers comparable accuracy with fewer features, improving the interpretability of sex classification.

We recorded importance–building times for each feature-selection method under a unified protocol. On the Markov CTh dataset, the human cohort required LIME: 288.57 seconds, SHAP: 447.71 seconds, and NEOFIT: 107.04 seconds to build feature importances; for the macaque cohort the corresponding times were 1064.00 seconds, 100.62 seconds, and 194.47 seconds. Times are the median over 5 stratified resamples, and all experiments were executed on a dual-socket Intel Xeon Gold 6248R system (2×24 physical cores, 96 hardware threads; base 3.0 GHz, turbo 4.0 GHz), using 50 parallel workers for both training and inference.

## Voxel-wise feature importance maps

We applied MORF to gray matter and white matter MRI volume data for both humans and macaques separately, generating feature importance values for each voxel along with their corresponding $p$-values. Fig 6 illustrates the $p$-values generated using our feature importance testing algorithm, NEOFIT (see Algorithm 1), while figures (S6–S9 Figs) displaying the corresponding raw feature importance values are provided in the supplementary material.

Regions with significant feature importance for sex classification in humans were identified primarily in the limbic system, including the amygdala, hippocampus, and thalamus, as well as in occipital regions (Fig 6). The involvement of the limbic structures, which are implicated in motor coordination, sensory integration, and emotion regulation [26], aligns with previous findings that link sexual dimorphism in humans to emotional regulation and memory processing, functions predominantly associated with the amygdala and hippocampus [3]. Similarly, the occipital regions, responsible for visual processing, have shown structural differences between sexes, potentially reflecting sex-specific adaptations in sensory processing [3].

In macaques, feature importance was also primarily concentrated in the limbic system, encompassing the superior temporal gyrus, dentate gyrus (part of hippocampus), putamen, and caudate nucleus (Fig 6). Additional contributions were observed in regions directly connected to the limbic system, such as the orbital gyrus, insula, and claustrum, and in regions with indirect interactions, including the precentral gyrus which is primarily associated with motor function (Fig 6).

## Parcel-wise feature importance maps

For the CTh data, we applied SPORF to the parcellated data, utilizing both the Schaefer and Markov parcellation schemes for humans and macaques. The feature importance maps, showing raw feature importance values across all parcels, as

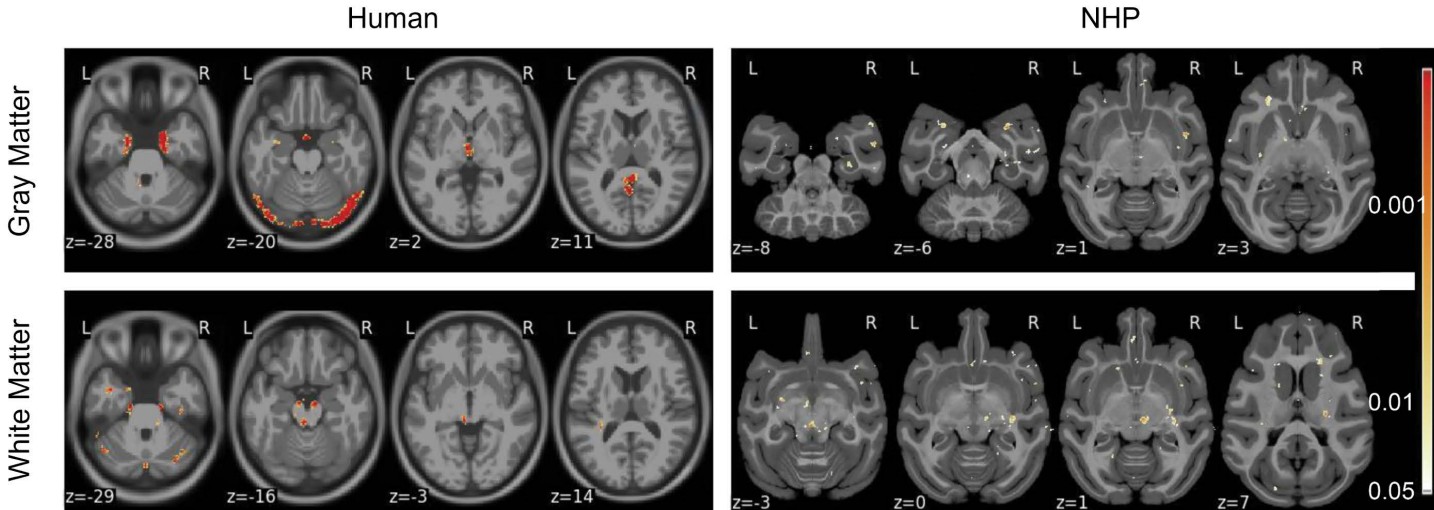

**Fig 6. Voxel-wise feature importance values and corrected *p*-values for human (left column) and NHP (right column).** Results are computed using our feature importance testing algorithm with $n_{permutations} = 50,000,000$. Selected horizontal slices of structural MRI volume on the MNI template show regions with at least 20 voxels and significant feature importance values ($p < 0.05$). All slices were visually examined, and only those with visibly large regions are presented here.

well as their corresponding *p*-values, are provided in the supporting information (S10–S13 Figs). To facilitate comparisons of significance patterns between the species, we merged the *p*-value maps for humans and macaques into a single surface, displayed with distinct color schemes (Fig 7). For parcels where both species exhibit significant *p*-values, we use a mosaic coloring pattern to highlight the significance levels for each species (Fig 7). This approach emphasizes regions where both species show consistent patterns of significance, as well as where differences exist.

The Markov parcellation reveals that human-specific regions are concentrated in networks associated with higher-order cognitive processes, including the dorsal and ventral attention networks and the default mode network (DMN). These cortical thickness differences likely reflect species-specific traits related to social cognition and complex behaviors. In contrast, macaque-specific regions overlap primarily with the posterior ventral attention network (Fig 7), indicating sex-related differences predominantly in sensory-driven or attentional systems. The orbitofrontal–limbic network emerges as a significant shared region across species (Fig 7), consistent with its conserved role in emotion and reward processing among primates.

For the Schaefer parcellation, shared significant regions are largely part of the posterior ventral attention network (Fig 7), underscoring the evolutionary conservation of attentional mechanisms underlying sex-related cortical differences. In humans, a broader array of networks, including somatomotor, frontoparietal, DMN, and visual networks, is implicated (Fig 7), highlighting the functional and structural complexity of human cortical organization. This diversity is likely linked to advanced motor control, cognition, and social interaction. Similar to the Markov parcellation, macaque-specific regions are confined to posterior networks, such as the posterior dorsal and ventral attention networks and the DMN (Fig 7), suggesting a more limited scope of sex-based cortical differences compared to humans.

We conducted a network-level analysis of the CTh data by aggregating parcels according to the 7-Yeo brain network scheme [27]. A Wilcoxon signed-rank test was applied to compare feature importances at the network level between humans and macaques across the seven Yeo networks: Visual, Somatomotor, Dorsal Attention, Ventral Attention, Limbic, Frontoparietal, and Default. The Frontoparietal network showed a significant difference in feature importances between species in the Schaefer parcellation, with humans exhibiting higher feature importance values (Fig 8).

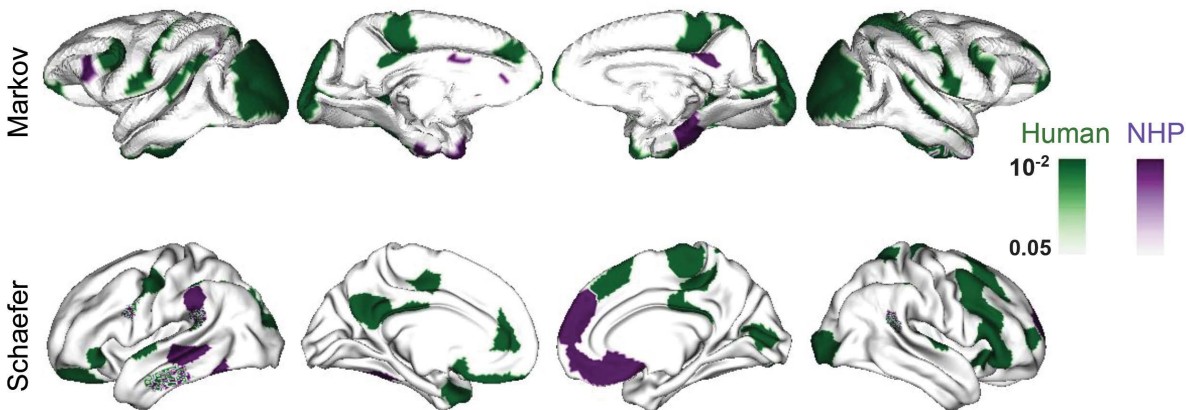

**Fig 7. Parcel-wise feature importance values and corrected *p*-values mapped onto brain surfaces for human (green) and NHP (purple).** Results are computed using our feature importance testing algorithm with $n_{permutations} = 5000$. Rows show parcellations: top – Markov, bottom – Schaefer. Significance is determined using a threshold of $p < 0.05$ for either species. Parcels where both species meet the significance threshold are highlighted with a mosaic pattern, indicating overlap of significant results while preserving species-specific contributions. This visualization enables direct comparison of significant brain regions across humans, non-human primates, or both.

## Discussion

Our study demonstrates that the integration of oblique random forests (ORFs) with permutation-based feature importance testing fills a critical gap in neuroimaging analysis. By capturing complex, non-linear interactions through oblique splits and rigorously validating feature importance, our method reliably distinguishes true signal from noise in high-dimensional data. This advancement is evident in our simulation studies, where our approach outperforms conventional methods such as LIME and SHAP, and is further validated in the classification of sex from both voxel-wise MRI and parcellated CTh data.

Our analysis estimated the overall effect of biological sex on brain structure, representing sex effects averaged across the age distributions observed in our sample. Under the causal framework outlined in (Fig 2), this can be conceptualized as an average treatment effect (ATE) [28]. Our human data exhibited an age imbalance between sexes (females: mean 39 years, males: mean 31 years) arising from aggregation of datasets with different demographic compositions. If sample inclusion depended jointly on both sex and age, conditioning on the selected sample could induce collider bias [29]. While we found no explicit evidence of such joint selection mechanisms in the data collection protocols, we cannot definitively rule out this possibility in large-scale multi-site aggregations.

Structural MRI volume and CTh data introduce distinct challenges for classification. Voxel-wise MRI data is inherently high-dimensional, leading to challenges such as feature redundancy, increased measurement noise, and the curse of dimensionality. While MORF addresses local feature dependencies by considering spatially correlated feature patches, distinguishing meaningful biological signals from noise remains difficult, particularly given the variability in MRI acquisition protocols across datasets.In contrast, CTh data is lower-dimensional but relies on template-driven parcellation schemes, which may not optimally capture fine-grained anatomical differences across individuals. Parcellation choices can constrain classification performance, as predefined regions may not align with the most relevant biological variations. This limitation can be exacerbated when parcellations are derived from standard population-based templates rather than subject-specific cortical features.

Our sex classification results provide important insights into the underlying neuroanatomical patterns. Specifically, the identification of both shared and species-specific regions suggests a balance between evolutionary conservation and species adaptation. In humans, significant regions—primarily within the limbic system and higher-order cognitive

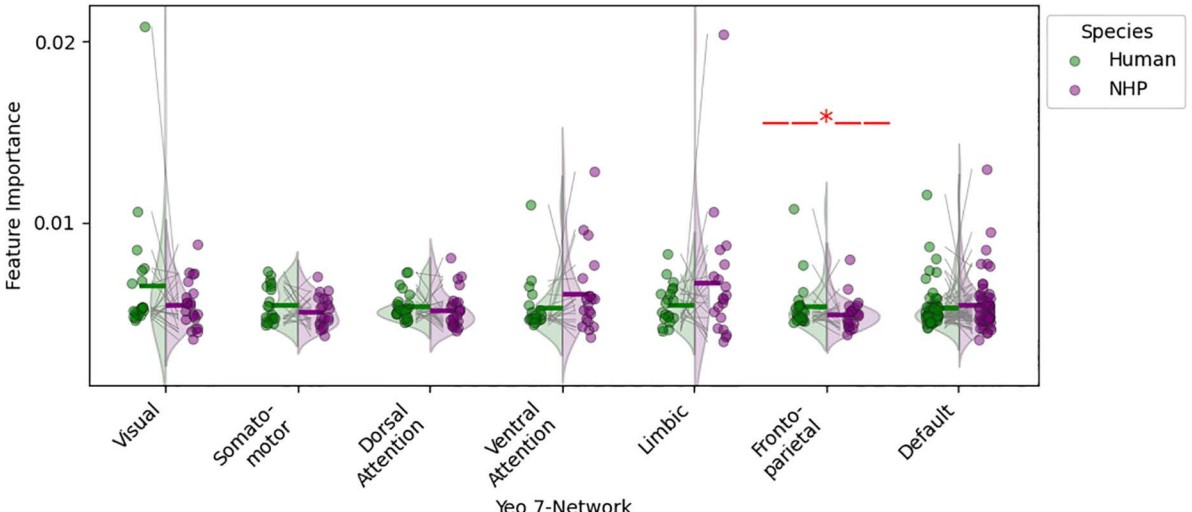

**Fig 8. Comparison of network-wise classification feature importances across species for the Schaefer parcellation (a similar plot for the Markov parcellation can be found in the supporting information S14 Fig).** For each species, the raw feature importance values for all parcels corresponding to the same network, according to Yeo's 7-network, were aggregated. The differences in feature importances across species were then evaluated using the Wilcoxon signed-rank test. The short horizontal line above each strip indicates the mean, and the red star highlights the pair with a significant difference ($p < 0.05$).

areas—underscore the role of emotion and memory processing in sex differentiation. Conversely, in macaques, regions linked to sensory processing and motor control appear more prominent. This dichotomy aligns with prior evidence of conserved neural mechanisms across primates while also reflecting species-specific adaptations [3,5,21]. While both species demonstrate significant sex-related differences in the limbic system, the specific regions implicated differ, reflecting species-specific neurobiological adaptations. Humans show pronounced involvement of the amygdala and hippocampus, emphasizing emotional and memory-related sex differences, whereas macaques exhibit a broader network involving motor, sensory, and integrative regions such as the precentral gyrus and insula. These findings may reflect evolutionary divergence in sex-specific neural functions, with humans demonstrating more specialization in cognitive and emotional domains. Collectively, our findings highlight the potential of our method to reveal meaningful structural markers that can inform our understanding of neurobiological sex differences and their evolutionary origins.

## Limitations

Despite these promising results, several limitations remain. First, although our permutation-based feature importance testing improves statistical validation, voxel-wise data—with a vast number of features—require an extremely high number of permutations to reliably correct $p$-values, resulting in significant computational cost. Second, our current classification accuracy for human MRI volume and cortical thickness data is moderate, which raises concerns regarding the robustness of the identified features as reliable markers. Finally, while our approach outperforms established methods like LIME and SHAP in our simulations, further validation on independent datasets and exploration of computational efficiency in extremely high-dimensional settings remain important avenues for future research.

## Future work

Building on the above limitations, we outline several directions for future work. First, incorporating additional MRI-derived features (e.g., cortical curvature and white-matter connectivity) could improve classification and yield a more

comprehensive view of sex-linked structural variation. Second, extending cross-species analyses to additional primates would help clarify evolutionary patterns and the extent to which observed differences are conserved across species. Third, transfer-learning approaches—such as training on human data and adapting to macaque data—may leverage shared structure while accounting for species-specific differences, thereby improving cross-species comparability. Furthermore, evaluating our method on a population-scale dataset such as the UK Biobank would provide a stringent test of out-of-sample generalization and scalability [31,32].

In addition, future work could examine conditional average treatment effects (CATEs) across age to assess whether sex differences vary across developmental stages and aging [24,28,30], providing a temporal perspective on sexual dimorphism while potentially mitigating selection-bias concerns introduced by conditioning on ICV. In the context of Fig 2, such an analysis would permit identification of causal effects [?].

We also plan to incorporate honest tree procedures (sample-splitting trees in which one subsample determines splits and a separate subsample estimates leaf predictions and related quantities), as proposed in [33,34]. Coupling NEOFIT with honest splitting is expected to reduce adaptive bias in importance estimates and improve the reliability of importance estimators.

To reduce NEOFIT build time, we leveraged Google's Yggdrasil Decision Forests (YDF; https://github.com/google/yggdrasil-decision-forests) to accelerate SPORF training, yielding substantial speedups on cortical-thickness (CTh) classification tasks. As a next step, we will integrate a YDF-backed ORF backend into the NEOFIT implementation to further reduce training and inference times.

Beyond fundamental structural differences, future work should examine sex-related brain variation in the context of neurological and psychiatric disorders, where sex is a known risk factor (e.g., autism, schizophrenia, depression). Understanding how these differences manifest in both healthy and clinical populations may refine diagnostic tools and inform personalized treatment strategies. More broadly, integrating machine learning with rigorous statistical validation has the potential to reshape neuroimaging analysis, paving the way for more interpretable and biologically meaningful discoveries in neuroscience.

## Supporting information

**S1 Appendix. Mathematical Formulation of the Trunk Simulation.**
(DOCX)

**S1 Table. List of human structural MRI volume datasets.**
(DOCX)

**S1 Fig. Flow diagram for the classification framework.**
(TIF)

**S2 Fig. Sex and age distributions in human and macaque MRI volume datasets.**
(TIF)

**S3 Fig. Sex and age distributions in human and macaque cortical thickness datasets.**
(TIF)

**S4 Fig. Performance comparison of oblique random forest (SPORF) on macaque cortical thickness data under different normalization methods, with n_estimators values set to 20,000 and max_features set to *p*.** The results indicate that data normalization improves performance across both parcellation schemes. Notably, Z-score normalization yields a substantial improvement in the Markov parcellation, whereas in the Schaefer parcellation, all three normalization methods exhibit comparable performance.
(TIF)

 

**S5 Fig. Comparison of feature selection performance on Schaefer parcellation data.**
(TIF)

**S6 Fig. Raw feature importance values for MRI volume human gray matter.**
(TIF)

**S7 Fig. Raw feature importance values for MRI volume macaque gray matter.**
(TIF)

**S8 Fig. Raw feature importance values for MRI volume human white matter.**
(TIF)

**S9 Fig. Raw feature importance values for MRI volume macaque white matter.**
(TIF)

**S10 Fig. Parcel-wise feature importances and corrected *p* values for human on Schaefer parcellation.**
(TIF)

**S11 Fig. Parcel-wise feature importances and *p* values for macaque on Schaefer parcellation.**
(TIF)

**S12 Fig. Parcel-wise feature importances and corrected *p* values for human on Markov parcellation.**
(TIF)

**S13 Fig. Parcel-wise feature importances and corrected *p* values for macaque on Markov parcellation.**
(TIF)

**S14 Fig. Comparison of network-wise classification feature importances across species for the Markov parcellation.** For each species, the raw feature importance values for all parcels corresponding to the same network, according to Yeo's 7-network, were aggregated. The differences in feature importances across species were then evaluated using the Wilcoxon signed-rank test. The short horizontal line above each strip indicates the mean.
(TIF)

## Code availability

The code used to perform the analysis and generate results for sex classification can be accessed at https://github.com/neurodata/sex_classification. NEOFIT implementation can be tracked at https://github.com/neurodata/treeple.

## Acknowledgments

The authors would like to thank Yuxin Bai and Haoyin Xu for their valuable insights and advice on this work.

## Author contributions

**Conceptualization:** Tingshan Liu, Jayanta Dey, Joshua T. Vogelstein.

**Data curation:** Tingshan Liu, Jayanta Dey, Samuel Alldritt, Karl-Heinz Nenning, Kyoungseob Byeon, Ting Xu.

**Formal analysis:** Tingshan Liu, Jayanta Dey, Beiya Xu.

**Funding acquisition:** Ting Xu, Joshua T. Vogelstein.

**Methodology:** Tingshan Liu, Jayanta Dey, Ting Xu, Joshua T. Vogelstein.

**Software:** Tingshan Liu, Jayanta Dey.

**Supervision:** Ting Xu, Joshua T. Vogelstein.

**Writing – original draft:** Tingshan Liu.

**Writing – review & editing:** Jayanta Dey, Eric W. Bridgeford, Ting Xu, Joshua T. Vogelstein.

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
