## [Decision Letter · Decision Letter 0]

2 Oct 2025

PONE-D-25-39004Statistically valid explainable black-box machine learning: applications in sex classification across species using brain imagingPLOS ONE

Dear Dr. Liu,

Thank you for submitting your manuscript to PLOS ONE. After careful consideration, we feel that it has merit but does not fully meet PLOS ONE’s publication criteria as it currently stands. Therefore, we invite you to submit a revised version of the manuscript that addresses the points raised during the review process.

We look forward to receiving your revised manuscript.

Kind regards,

Ravi Bansal, Ph.D.

Academic Editor

PLOS ONE

Journal Requirements:

2. Please note that PLOS One has specific guidelines on code sharing for submissions in which author-generated code underpins the findings in the manuscript. In these cases, we expect all author-generated code to be made available without restrictions upon publication of the work.

Please review our guidelines at https://journals.plos.org/plosone/s/materials-and-software-sharing#loc-sharing-code and ensure that your code is shared in a way that follows best practice and facilitates reproducibility and reuse.

**Additional Editor Comments:**

Dear Authors:

Please see attached the comments from the reviewers. I agree with the reviewers that the performance of your proposed method should be compared with other machine learning algorithms. And that your procedures needs to be detailed better for ease of understanding.

I am therefore glad to offer reconsideration of your manuscript after you have addressed the reviewers' concerns.

Kindly submit a revised manuscript with and without track changes and a cover letter detailing how each of the reviewer concern was addressed in the revised manuscript.

Sincerely,

Ravi Bansal, Ph.D.

Reviewers' comments:

Reviewer's Responses to Questions

**Comments to the Author**

1. Is the manuscript technically sound, and do the data support the conclusions?

Reviewer #1: Yes

Reviewer #2: Yes

2. Has the statistical analysis been performed appropriately and rigorously? 

Reviewer #1: Yes

Reviewer #2: Yes

3. Have the authors made all data underlying the findings in their manuscript fully available?

The PLOS Data policy requires authors to make all data underlying the findings described in their manuscript fully available without restriction, with rare exception (please refer to the Data Availability Statement in the manuscript PDF file). The data should be provided as part of the manuscript or its supporting information, or deposited to a public repository. For example, in addition to summary statistics, the data points behind means, medians and variance measures should be available. If there are restrictions on publicly sharing data—e.g. participant privacy or use of data from a third party—those must be specified.requires authors to make all data underlying the findings described in their manuscript fully available without restriction, with rare exception (please refer to the Data Availability Statement in the manuscript PDF file). The data should be provided as part of the manuscript or its supporting information, or deposited to a public repository. For example, in addition to summary statistics, the data points behind means, medians and variance measures should be available. If there are restrictions on publicly sharing data—e.g. participant privacy or use of data from a third party—those must be specified.requires authors to make all data underlying the findings described in their manuscript fully available without restriction, with rare exception (please refer to the Data Availability Statement in the manuscript PDF file). The data should be provided as part of the manuscript or its supporting information, or deposited to a public repository. For example, in addition to summary statistics, the data points behind means, medians and variance measures should be available. If there are restrictions on publicly sharing data—e.g. participant privacy or use of data from a third party—those must be specified.requires authors to make all data underlying the findings described in their manuscript fully available without restriction, with rare exception (please refer to the Data Availability Statement in the manuscript PDF file). The data should be provided as part of the manuscript or its supporting information, or deposited to a public repository. For example, in addition to summary statistics, the data points behind means, medians and variance measures should be available. If there are restrictions on publicly sharing data—e.g. participant privacy or use of data from a third party—those must be specified.

Reviewer #1: Yes

Reviewer #2: Yes

4. Is the manuscript presented in an intelligible fashion and written in standard English?

Reviewer #1: Yes

Reviewer #2: Yes

5. Review Comments to the Author

Reviewer #1: Could you provide a quantitative comparison of your ORF+NEOFIT framework against other state-of-the-art classifiers standard Random Forests, SVMs, or simple neural networks and interpretability methods (LIME, SHAP) in terms of both classification accuracy and the reliability of feature importance scores on your datasets?

What were the key performance metrics Accuracy, AUC, F1-Score achieved by your model on the human sMRI and cortical thickness datasets, as well as on the macaque data?

Beyond capturing intricate interactions, what is the specific neuroscientific or data-driven rationale for believing that oblique (linear) decision boundaries are more appropriate than axis-aligned ones for modeling the relationship between voxels/cortical features and sex?

Please elaborate on the NEOFIT algorithm. Specifically, how is the null distribution for each feature's importance score generated permuting labels, permuting features? What specific procedure is used to correct the p-values for multiple comparisons across thousands of features?

The results promise "interpretable insights." Can you provide examples of the top neuroanatomical features identified by NEOFIT and discuss whether they align with previously established sex differences in the literature in the limbic system, cortex thickness patterns?

A key claim is the facilitation of cross-species comparisons. Were there any notable similarities or differences in the important features identified between humans and macaques? What might the evolutionary implications of these findings be?

Could you provide details on the human and macaque datasets used? This includes the number of subjects, age range, sex distribution, and scanner protocols, as these factors can significantly influence the results.

Sex differences in brain structure are known to be influenced by factors like age, brain size (ICV), and hormonal status. How were these potential confounders controlled for in your analysis, either during data preprocessing or within the model itself?

Highlighted article might be considered for related work section. (https://doi.org/10.1016/j.compeleceng.2022.108405

)

ORFs and permutation-based testing are computationally intensive. Could you comment on the training and inference time of your framework compared to traditional methods, and its scalability to even larger datasets UK Biobank?

Will the code for the integrated ORF+NEOFIT framework be made publicly available? This is crucial for the adoption of the method by the wider neuroscience community.

Reviewer #2: - The research idea and writing were good.

- It's best to include a related works on this topic and compare the proposed work with previous results.

- The number of samples taken from each dataset is assumed to be mentioned.

- Mention the results obtained in the abstract.

6. PLOS authors have the option to publish the peer review history of their article (what does this mean?). If published, this will include your full peer review and any attached files.). If published, this will include your full peer review and any attached files.). If published, this will include your full peer review and any attached files.). If published, this will include your full peer review and any attached files.

...

Reviewer #1: No

Reviewer #2: No

---

## [Author Response · Author response to Decision Letter 1]

20 Feb 2026

Statistically valid explainable black-box machine learning: applications in sex classification across species using brain imaging (PLOS One)

REVIEWER #1

1. Reviewer: Could you provide a quantitative comparison of your ORF+NEOFIT framework against other state-of-the-art classifiers standard Random Forests, SVMs, or simple neural networks and interpretability methods (LIME, SHAP) in terms of both classification accuracy and the reliability of feature importance scores on your datasets?

Authors: Thank you for bringing this question to the table. This is an important question and added to our manuscript’s results. For classification performance, we have added Random Forest (RF), support vector machine (SVM), and a simple multilayer perceptron (MLP) as baselines for the cortical thickness (CTh) task and report classification accuracy as the primary metric. Details of each model for classification are provided as follows: In SPORF, we used the hyperparameters determined in the Hyperparameter Tuning section. For parity, the number of estimators was set to 5,000 for human data and 20,000 for macaque data to match the same tree counts in SPORF, and SVM used an RBF kernel. The MLP was implemented using MLPClassifier in scikit-learn, configured with two hidden layers (256, 256). All methods use identical preprocessing (normalization) and the same fixed train-validation-test splits. Rebuttal Fig 1 shows the learning curves of each model.

We notice that in human data, SPORF consistently outperforms axis-aligned RF and remains competitive with SVM and MLP, showing the most monotonic increase of accuracy as data increases. In macaque data, where sample size is smaller and heterogeneity higher, variance grows for all methods, yet SPORF retains a stable edge over RF. The pattern is consistent across parcellations.

For feature-selection performance on CTh, we used a common RF backbone for all post-hoc methods. Specifically, we applied LIME and TreeSHAP to RF and also evaluated Gini-impurity feature importances from the scikit-learn RF. To assess the reliability of each explanation method, we re-train each classifier by using only the significant features selected by that method and report the resulting accuracy, see lines 352-363 and Fig 5 in the Results section. Code and scripts to reproduce all figures are available at the GitHub repository cited in the Data and code availability section.

Due to the large scale of voxel-wise volume data, baseline methods would presently require substantial additional computation under a fair protocol (hyperparameter optimization, compute-matched evaluation, etc.), description is added to Discussion - Limitations subsection, lines 489-492. Therefore, experiments on MRI data are not included in this revision. We have queued voxel-wise baselines as a planned extension (see Discussion - Future Work subsection, lines 500-503).

2. Reviewer: What were the key performance metrics Accuracy, AUC, F1-Score achieved by your model on the human sMRI and cortical thickness datasets, as well as on the macaque data?

Authors: Thank you for your comment for adding performance metrics. In the Results section, Fig 3 presents out-of-bag (OOB) accuracies for SPORF on cortical thickness (CTh) data across the feature_combinations and max_features hyper-parameters, with peaks of about 0.74 in humans and around 0.66 in macaques, consistent across both parcellations.

Fig 4 shows ROC curves for the eight experiments. AUCs are reported for ORFs in both parcellations, human volume ≈ 0.97, human CTh = 0.81, macaque volume = 0.73, and macaque CTh ≈ 0.69. The statement can be found between line 344-351.

We agree that F1 Score can be informative and reliable, especially under substantial class imbalance. In our data, the sex proportions are near balanced in both species, so AUC with OOB accuracy could capture the main performance characteristics without introducing a thresholding choice.

3. Reviewer: Beyond capturing intricate interactions, what is the specific neuroscientific or data-driven rationale for believing that oblique (linear) decision boundaries are more appropriate than axis-aligned ones for modeling the relationship between voxels/cortical features and sex?

Authors: Thank you for raising this important question. This is one of the key points for ORFs. To explain this better, we introduce Figure 4 in Sparse Projection Oblique Randomer Forests from Tomita et al. as our Rebuttal Fig 2.

Rebuttal Fig 2A & B show that the top split-node projections in SPORF are linear combinations of dimensions, whereas RF is restricted to single dimensions. These SPORF projections preferentially load on the early, signal-bearing dimensions. As summarized in Rebuttal Fig 2C & D, the normalized Gini importances of SPORF’s top projections exceed those of RF (except the first RF feature), and Bayes error rates of SPORF’s projects are much lower than RF’s, indicating that SPORF learns more informative, identifiable features, while RF is limited to create new features beyond axis-aligned thresholds.

In conclusion, in high-dimensional, correlated voxel/cortical feature spaces, oblique boundaries offer a principled and empirically supported way to capture network-level and distributed effects that are expected in sex classification using MRI structural data.

4. Reviewer: Please elaborate on the NEOFIT algorithm. Specifically, how is the null distribution for each feature's importance score generated permuting labels, permuting features? What specific procedure is used to correct the p-values for multiple comparisons across thousands of features?

Authors: Thank you for the opportunity to clarify NEOFIT; we apologize for any confusion. The procedure can be described as follows: (i) For each bootstrap sample, we fit two trees on identical inputs, one with the original labels (observed importances) and one with randomly permuted labels (null importances). In this step, we do not permute features, but only permute labels. (ii) For each feature, we compute a paired statistic that records how often the null importance exceeds the observed across trees. (iii) We construct the null distribution by repeatedly and randomly swapping the “observed/null” labels within each tree pair’s importance vectors and recomputing the statistic many times. (iv) The feature-level p-value is the proportion of permuted statistics that exceed the observed statistic.

To address multiple comparisons across thousands of features, we apply the Holm-Bonferroni correction to the feature-wise p-values and report adjusted p-values, declaring significance based on these.

Full pseudocode is provided as Algorithm 1 in the Materials and methods - Feature importance testing subsection. And the description can be found between lines 115-129 in the same section.

5. Reviewer: The results promise "interpretable insights." Can you provide examples of the top neuroanatomical features identified by NEOFIT and discuss whether they align with previously established sex differences in the literature in the limbic system, cortex thickness patterns?

Authors: Thank you for this question. The statistically significant features selected by NEOFIT are concentrated in neuroanatomically plausible systems.

In humans, we observe limbic structures (amygdala, hippocampus including dentate gyrus, thalamus) and occipital/visual cortex among the highest-ranking regions (see Fig 6), with cortical thickness effects spanning dorsal and ventral attention and default-mode networks under the Markov parcellation and extending to somatomotor, frontoparietal, default-mode, and visual networks under the Schaefer parcellation (see Fig 7). These patterns align with prior reports of sex differences in affective and memory-related circuitry, as well as in visual cortex.

In macaques, significant features cluster in limbic and basal-ganglia territories (superior temporal gyrus, dentate gyrus, putamen, caudate nucleus) and in regions connected to limbic circuitry (orbitofrontal gyrus, insula, claustrum, precentral gyrus), with a predominance in posterior dorsal and ventral attention networks and the default-mode network under the parcel-wise analyses (Fig 7), indicating a stronger sensory–attentional emphasis. Cross-species, the orbitofrontal–limbic network emerges as a shared substrate (Fig 7).

At the network level, aggregating cortical thickness feature importances according to Yeo’s 7-network parcellation shows higher frontoparietal importance in humans under the Schaefer scheme, consistent with more extensive higher-order control systems in humans. Together, these results provide concrete, interpretable features that are consistent with established sex-dimorphic neuroanatomy while highlighting species-specific emphases.

6. Reviewer: A key claim is the facilitation of cross-species comparisons. Were there any notable similarities or differences in the important features identified between humans and macaques? What might the evolutionary implications of these findings be?

Authors: We really appreciate this question, and we made it more clearer in our current submission. In our work, we observed both conserved and species-specific patterns among NEOFIT-identified features. Shared signal concentrated in limbic circuitry (amygdala/hippocampus and orbitofrontal–limbic regions), consistent across Markov and Schaefer parcellations and aligned with prior reports of sex-linked differences in emotion/reward systems in primates. Human-specific emphasis appeared in association cortex—frontoparietal, default-mode, and attention networks—suggesting that sex-related effects in humans involve distributed higher-order systems. Macaque-specific emphasis was more posterior and sensorimotor/attentional (e.g., precentral gyrus, insula; posterior ventral attention network). A network-level summary (Yeo-7) echoed this pattern, with stronger frontoparietal weighting in humans and relatively greater posterior attention/sensorimotor weighting in macaques. Detailed description can be found in the Results section, specifically in the Voxel-wise feature importance maps and Parcel-wise feature importance maps subsection

8. Reviewer: Could you provide details on the human and macaque datasets used? This includes the number of subjects, age range, sex distribution, and scanner protocols, as these factors can significantly influence the results.

Authors: Thank you for raising this very important point to consider. For MRI volume data, we described details of age and sex distributions between lines 231-240 in Materials and methods - Structural MRI dataset subsection. For cortical thickness data, we mentioned the distributions between lines 293-294 in Materials and methods - Cortical thickness dataset subsection.

In order to characterize the sex and age distributions better, we also added additional histograms with small bin widths. See S4 Fig and S5 Fig in Supplementary Information section.

Information for scanner protocols of each dataset can be found in each dataset’s original paper. Every work for the dataset has been cited in S2 Table in the Supplementary Information section.

9. Reviewer: Sex differences in brain structure are known to be influenced by factors like age, brain size (ICV), and hormonal status. How were these potential confounders controlled for in your analysis, either during data preprocessing or within the model itself?

Authors: Thank you for the comment, these factors are very important to be considered. Age, intracranial volume (ICV), and hormonal status can indeed act as confounders in sex-related analyses. In our original submission, we did not adjust for these factors during preprocessing or within our classifiers, in order to present an initial benchmark of ORF+NEOFIT on the raw imaging features.

We added a new subsection titled Variable specification and relationship identification for within and cross-species MRI analysis under Materials and methods section (lines 295-317) and an accompanying directed acyclic graph (Fig. 2) describing the assumed causal structure for sex classification analysis.

We acknowledge this as a limitation of our present study and have outlined concrete robustness checks for subsequent analyses in the future between lines 511-514 in the Discussion - Future Work.

10. Reviewer: Highlighted article might be considered for related work section. (https://doi.org/10.1016/j.compeleceng.2022.108405)

Authors: Thank you for highlighting this excellent work. We have incorporated it into the Introduction section as an introductory sentence in lines 3-5 to contextualize our study. We believe this addition strengthens the background and situates our contribution within evidence of strong predictive performance from ML-based pipelines in medical imaging, while motivating our structural MRI focus on pairing accuracy with statistically validated, interpretable feature importance for sex classification.

11. Reviewer: ORFs and permutation-based testing are computationally intensive. Could you comment on the training and inference time of your framework compared to traditional methods, and its scalability to even larger datasets UK Biobank?

Authors: This comment raises a very crucial point to consider. We have addressed this question by running NEOFIT and baseline methods on CTh for both species, then recording the training time and inference time of each method. Time results are reported between lines 372-379 in the Results Section.

We did not add a UK Biobank experiment at this time, since it would require substantial additional resources for very long-time experiments. However we note that our classifier accepts all kinds of data, so it is compatible with UK Biobank tasks. We agree that including such a public large dataset is a crucial step to validate the generalization ability for our ORFs and we have added this to Discussion - Future Work between lines 508-510 as a planned generalization experiment.

12. Reviewer: Will the code for the integrated ORF+NEOFIT framework be made publicly available? This is crucial for the adoption of the method by the wider neuroscience community.

Authors: Thank you very much for this comment. We share PLOS One’s commitment to open science. The full code for the integrated ORF+NEOFIT framework and all the analysis/experiments in our paper is publicly available and the link of the code is provided in the newly added Data and code availability section in lines 571-577. The GitHub repository includes primary analysis, training scripts and figure generation for reproducibility. We also expose a callable class, Neuro-Explainable Optimal Feature Importance Testing (NEOFIT), via the treeple package. This enables NEOFIT with ORF backbones for feature-importance inference across datasets. Installation guidance and examples are documented in Pull Request #356 in the treeple repository. We sincerely expect this work will provide useful insights to the neuroscience community.

REVIEWER #2

1. Reviewer: It's best to include related works on this topic and compare the proposed work with previous results.

Authors: Thank you for emphasizing the importance of situating our work within prior results. Our primary aim in this paper is to establish a statistically valid, interpretable pipeline for high-dimensional neuroimaging; accordingly, in this revision we prioritized focused updates over an exhaustive survey. Specifically, we added concise references to ML-based diagnostic pipelines in radiography and ultrasound to motivate the broader imag

---

## [Decision Letter · Decision Letter 1]

23 Mar 2026

Statistically valid explainable black-box machine learning: applications in sex classification across species using brain imaging

PONE-D-25-39004R1

Dear Dr. Liu,

We’re pleased to inform you that your manuscript has been judged scientifically suitable for publication and will be formally accepted for publication once it meets all outstanding technical requirements.

Kind regards,

Ravi Bansal, Ph.D.

Academic Editor

PLOS One

Additional Editor Comments (optional):

Reviewers' comments:

Reviewer's Responses to Questions

**Comments to the Author**

1. If the authors have adequately addressed your comments raised in a previous round of review and you feel that this manuscript is now acceptable for publication, you may indicate that here to bypass the “Comments to the Author” section, enter your conflict of interest statement in the “Confidential to Editor” section, and submit your "Accept" recommendation.

Reviewer #1: All comments have been addressed

2. Is the manuscript technically sound, and do the data support the conclusions?

Reviewer #1: Yes

3. Has the statistical analysis been performed appropriately and rigorously? 

Reviewer #1: Yes

4. Have the authors made all data underlying the findings in their manuscript fully available?

The PLOS Data policy requires authors to make all data underlying the findings described in their manuscript fully available without restriction, with rare exception (please refer to the Data Availability Statement in the manuscript PDF file). The data should be provided as part of the manuscript or its supporting information, or deposited to a public repository. For example, in addition to summary statistics, the data points behind means, medians and variance measures should be available. If there are restrictions on publicly sharing data—e.g. participant privacy or use of data from a third party—those must be specified.requires authors to make all data underlying the findings described in their manuscript fully available without restriction, with rare exception (please refer to the Data Availability Statement in the manuscript PDF file). The data should be provided as part of the manuscript or its supporting information, or deposited to a public repository. For example, in addition to summary statistics, the data points behind means, medians and variance measures should be available. If there are restrictions on publicly sharing data—e.g. participant privacy or use of data from a third party—those must be specified.requires authors to make all data underlying the findings described in their manuscript fully available without restriction, with rare exception (please refer to the Data Availability Statement in the manuscript PDF file). The data should be provided as part of the manuscript or its supporting information, or deposited to a public repository. For example, in addition to summary statistics, the data points behind means, medians and variance measures should be available. If there are restrictions on publicly sharing data—e.g. participant privacy or use of data from a third party—those must be specified.requires authors to make all data underlying the findings described in their manuscript fully available without restriction, with rare exception (please refer to the Data Availability Statement in the manuscript PDF file). The data should be provided as part of the manuscript or its supporting information, or deposited to a public repository. For example, in addition to summary statistics, the data points behind means, medians and variance measures should be available. If there are restrictions on publicly sharing data—e.g. participant privacy or use of data from a third party—those must be specified.

Reviewer #1: Yes

5. Is the manuscript presented in an intelligible fashion and written in standard English?

Reviewer #1: Yes

6. Review Comments to the Author

Reviewer #1: The revised article have incorporated all suggested review comments in the updated manuscript. It might be considered for further process.

7. PLOS authors have the option to publish the peer review history of their article (what does this mean?). If published, this will include your full peer review and any attached files.). If published, this will include your full peer review and any attached files.). If published, this will include your full peer review and any attached files.). If published, this will include your full peer review and any attached files.

...

Reviewer #1: No

---

## [Editor Report · Acceptance letter]

PONE-D-25-39004R1

PLOS One

Dear Dr. Liu,

I'm pleased to inform you that your manuscript has been deemed suitable for publication in PLOS One. Congratulations! Your manuscript is now being handed over to our production team.

Kind regards,

on behalf of

Dr. Ravi Bansal

Academic Editor

PLOS One